# Quantized current steps due to the synchronization of microwaves with Bloch oscillations in small Josephson junctions

Rais S. Shaikhaidarov [1,2], Kyung Ho Kim[1], Jacob Dunstan[1], Ilya Antonov[1,2], Dmitry Golubev [3,4], Vladimir N. Antonov [1] ✉ & Oleg V. Astafiev [1,5]

Synchronization of Bloch oscillations in small Josephson junctions (JJs) under microwave radiation, which leads to current quantization, has been proposed as an effect that is dual to the appearance of Shapiro steps. This current quantization was recently demonstrated in superconducting nanowires in a compact high-impedance environment. Direct observation of current quantization in JJs would confirm the synchronization of Bloch oscillations with microwaves and help with the realisation of the metrological current standard. Here, we place JJs in a high-impedance environment and demonstrate dual Shapiro steps for frequencies up to 24 GHz ($I = 7.7$ nA). Current quantization exists, however, only in a narrow range of JJ parameters. We carry out a systematic study to explain this by invoking the model of a JJ in the presence of thermal noise. The findings are important for fundamental physics and application in quantum metrology.

A seminal paper by Likharev and Averin published in 1985 predicted that synchronization of Bloch oscillations in a Josephson junction (JJ) with microwaves (MWs) should result in quantized current steps, $I_m = 2efm$, where $e$ is the electron charge, $f$ is the microwave frequency, $m$ is an integer[1]. The effect is dual to the well-known quantization of voltage observed in the JJ, so-called the Shapiro steps. Significant efforts were made to observe this synchronization, with the most successful work performed by Kuzmin and Haviland who observed peaks in $dI/dV$ curves at $I = \pm 2ef$[2]. However, the demonstration of true current quantization was not achieved until decades later, when dual Shapiro steps were seen in superconducting nanowires[3]. This breakthrough experiment was inspired by the theoretical work of Nazarov and Mooij[4] and the observation of the coherent quantum phase slip (CQPS) through qubit spectroscopy[5].

Yet, the accuracy of the quantized steps in nanowires is inferior to that of alternative systems that facilitate a controllable classical charge transfer through a quantum dot[6] or turnstile pumps[7]. A significant drawback of nanowire devices is also their low fabrication yield, which is less than 30%. This comes as a result of their tiny, $\sim 10$ nm,

dimensions being at the limit of modern nanotechnology capabilities. Even nanowires of identical geometry have wide variations of the superconducting parameters, like critical current, and consequently, the CQPS energy[8]. Exploring current quantization in more reliable JJs through the synchronization of external MWs with the Bloch oscillations could provide a solution.

We note that steps were recently reported in ultra-small JJs in a high impedance environment composed of large JJs[9]. However, the synchronization of the Bloch oscillations (the modulation of the current steps with the MW amplitude), must be proven. We also note that the steps in nanowires discussed in ref. 10 are unlikely to be attributable to superconducting behaviour as it is pointed out in Ref. 11.

In this work, we focus on the experimental study of dual Shapiro steps in JJs. Different aspects of such a system have been previously studied theoretically[12–14]. It is important to find parameters of the JJs and the surrounding circuit that will protect the Bloch bands from external current noise while simultaneously allowing the coupling of MWs to the JJ. The Bloch bands are formed by the periodic modulation of the system's energy with the induced charge $q/2e$, see the central

[1]Royal Holloway University of London, Egham, Surrey TW20 0EX, UK. [2]National Physical Laboratory, Hampton Road, Teddington TW11 0LW, UK. [3]HQS Quantum Simulations GmbH, Rintheimer Str. 23, Karlsruhe 76131, Germany. [4]Department of Applied Physics, QTF Centre of Excellence, Aalto 610101, Finland. [5]Skolkovo Institute of Science and Technology, Bolshoy Boulevard 30, Moscow 121205, Russia. ✉e-mail: v.antonov@rhul.ac.uk

panel of Fig. 1a. The amplitude of this modulation, shown on the right panel of the figure, is the CQPS energy, $E_S$. The system's energy also oscillates with the superconducting flux, $\Phi/\Phi_0$. The energies in the graphs are normalised by the JJ energy, $E_J = \Delta R_Q/2R_N$, where $R_Q = h/4e^2 \approx 6.5\,k\Omega$ and $R_N$ are the quantum resistance and normal resistance of the JJ, $\Delta$ is the superconducting gap. The quantization of current occurs when the coherent tunnelling of the system with $E(q/2e)$ is synchronized with the external MWs.

The greatest challenge is to tackle Landau-Zener excitation between the lowest and the higher Bloch bands, which facilitates frequent switching between the branches of a hysteretic $I-V$ curve[15,16]. In response to this challenge, we explore parameters of the JJ and environmental circuit to find a balance between maximising $E_S$ and achieving adequate separation of the Bloch bands. At the same time, the JJ must have a differential quasiparticle resistance at low bias (below the superconducting gap $2\Delta/e$) comparable to the $R_Q$, and its Josephson energy close to the charging energy $E_J \sim E_C$[1,17]. Additionally, we protect the JJ from the EM noise of the environment by embedding it in an electric circuit with inductances, normal resistors, and quasiparticle traps (discussed in Supplementary Notes 3).

We experimentally demonstrate dual Shapiro steps, i.e., current quantization, in small JJs. The current plateaus are modulated with the MW amplitude, thus confirming the quantum coherent nature of the effect. We carefully explore the range of JJ parameters to maximize the quantized current and improve the accuracy of the quantization. Over 20 devices with different parameters of the JJ and sensitivity to the external MWs are explored. The largest observed quantized current and resistance at the current plateau are $I_1 \sim 7.7\,nA$ ($\sim 2e \times 23.895\,GHz$) and $3.4\,k\Omega$ respectively. The quantization is seen only in devices with a JJ area smaller than $100 \times 100\,nm^2$ and having an apparent critical voltage $V_C^* < 7\,\mu V$. The performance of the different devices has been analysed to determine the optimum JJ parameters.

A similar demonstration of the synchronization of Bloch oscillations with the MW and the current quantization in small Josephson Junctions was reported recently by the PTB group[18,19].

## Results & discussion

A helium FIB image of the JJ with the protective environment is shown in Fig. 1b. The JJ has lateral sizes 40 nm × 80 nm, which, together with Al pads of size $1 \times 1\,\mu m^2$ (light areas on the right panel before the dark TiN meander in Fig. 1b), give a capacitance $C_J \approx 0.3\,fF$, corresponding to a charging energy $E_{CJ} = e^2/2C_J \approx h \times 65\,GHz$. We found from the simulation of the experimental data that there is an additional stray capacitance $C_S \approx 1.2\,fF$ between the junction and metallic parts of the circuit. This reduces the charging energy to $E_C = e^2/2(C_J + C_S) \approx h \times 11\,GHz$. In what follows, we distinguish $E_{CJ}$ and $E_C$. The Josephson energy of the junction, calculated from its normal resistance ($R_N \approx 1.9\,k\Omega$), is $E_J = h \times 84\,GHz$, with $\Delta = h \times 49\,GHz$ being the superconducting gap of aluminium. Importantly, the corresponding plasma frequency is $E_p = \sqrt{8E_J E_C} \approx h \times 86\,GHz$. Thus, the upper energy band is separated from the ground band by a large energy gap, close to $2\Delta$, which reduces the Landau-Zener tunnelling. The ratio of the Josephson energy to the charging energy in the sample discussed below is $E_J/E_C = 7.6$.

The measurement circuit is shown on the right side of Fig. 1b. The JJ is embedded in a high impedance electromagnetic environment: there are four inductances arranged in sequence: two $L_1 = 1.15\,\mu H$ and two $L_2 = 0.34\,\mu H$, and four Pd resistors of $R = 6.3\,k\Omega$ screening the JJ from the external circuitry. The Pd/Al pads on both sides of $L_1$ serve as the quasiparticle traps. A *dc* four-probe measurement scheme is connected with current, $I+$ and $I-$, and voltage, $V+$ and $V-$, leads passed through a box with a cascade of LTCC low-pass filters with a stop band from 80 MHz to 20 GHz. The filter box is positioned at the 15 mK stage of the refrigerator. The screening circuit frequency band is

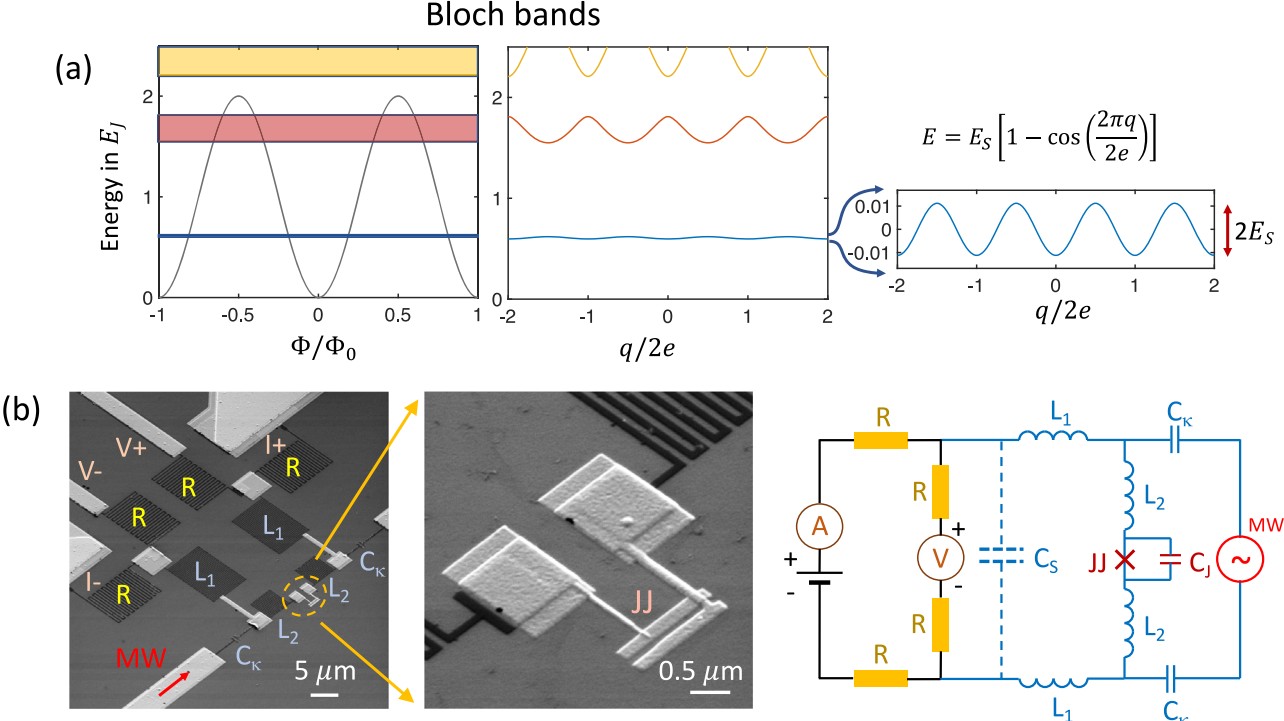

## Bloch bands

$$E = E_S\left[1 - \cos\left(\frac{2\pi q}{2e}\right)\right]$$

**Fig. 1 | Design and electric circuit of the sample and formation of the Bloch bands in JJ. a** The Bloch bands of the JJ in units of the Josephson energy $E_J$ versus normalized flux $\Phi/\Phi_0$ and charge $q/2e$ ($E_J/E_C = 4.4$). The lowest band shows $2e$ periodic oscillations in the charge space, with the amplitude corresponding to the phase-slip energy $E_S$. The three lowest Bloch bands are marked by different colours.

For current quantization the Bloch oscillations in the lowest band are synchronized with the external MWs; (**b**) He FIB image of the device and the equivalent electric circuit. A small JJ is embedded into a high impedance environment formed by TiN inductances $L_1 = 1.15\,\mu H$ and $L_2 = 0.34\,\mu H$ and normal Pd resistors $R = 6.3\,k\Omega$.

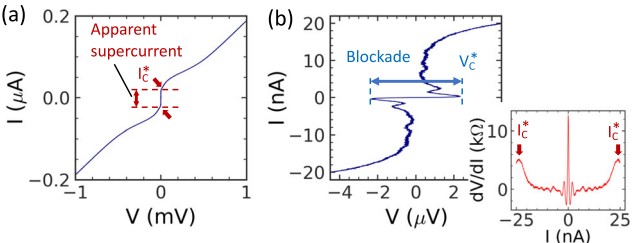

**Fig. 2 | Stationary $I$-$V$ curves of the JJ. a** $I-V$ curve in wide voltage range has supercurrent-like shape with an apparent supercurrent $I_C^* = 24$ nA. **b** $I-V$ curve in a narrower voltage range reveals a current blockade region with the apparent critical voltage $V_C^* = 2.33$ µV. Inset: a differential resistance $dV/dI$. The center peak is due to the current blockade. The two side maxima indicated by red arrows define the apparent current $I_C^*$.

$\Delta f_c = R/(2\pi L) = 0.6$ GHz. The MWs are coupled to the JJ by a capacitor with $C_\kappa = 0.1$ fF. The capacitance makes the highest impedance for the MW at the operation frequency, ~30 kΩ at 5 GHz, while the inductance $L_2$ between the $C_\kappa$ and the JJ has only ~100 Ω at this frequency. The role of $L_2$ is to screen the JJ from the noise of higher frequencies.

A stationary $I-V$ curve of the sample is shown in Fig. 2. In a wide voltage range, the curve shows supercurrent-like behaviour with the apparent critical current $I_C^* \approx 24$ nA (Fig. 2a). However, in a narrow voltage range (Fig. 2b) a current blockade appears. The current blockade up to the apparent critical voltage $V_C^* = 2.33$ µV is followed by a rise in current and voltage re-trapping[1,20]. The voltage does not go to zero but varies around 0.4 µV before returning to the usual branch of normal current above $I_C^*$. The curve is of great interest in itself due to its unusual properties: coexistence of the apparent supercurrent together with the current blockade at low voltage bias. The excess current of the $I-V$ curve is relatively small compared to that in CQPS devices with nanowires[3]. The bottom inset shows the differential resistance $dV/dI$, with the arrows indicating $I_C^*$.

The apparent critical current $I_C^*$ is likely determined by the Landau-Zener tunnelling current

$$I_z = \frac{\pi E_J}{16 E_{CJ}} I_C \qquad (1)$$

when the transitions from the lower to higher Bloch energy bands have a high probability ($I_C^* \leq I_Z$). Here, we take $E_{CJ}$ because for the high frequency, $f \sim I_C/2e$, the inductance provides a high impedance and effectively isolates the junction from the rest of the circuit. From the experiment, we have a ratio $I_C^*/I_C \sim 0.25$, which is consistent with Eq. 1 when taking $I_C = \pi\Delta/2eR_J = 170$ nA. The analysis is valid as long as $E_J < 2\Delta$.

The $I-V$ curve drastically changes when microwave radiation is applied to the JJ: current steps appear at $I_m = 2efm$. Examples of experimental curves taken at two frequencies, 6.495 GHz and 10.215 GHz, are shown in Fig. 3. Under the influence of 6.495 GHz MWs, current plateaus at $m=1$ and $m=2$ are developed with a low MW amplitude of $I_{ac} = 4.2$ nA, while the plateau at $m=3$ is seen when $I_{ac} = 5.8$ nA. There is only one clear quantized plateau under 10.215 GHz MWs. At both frequencies, the quantization is limited by the amplitude of the $I_C^*$. We explore JJs with different $E_J$ and $E_C$, and find that the necessary condition for quantization is $E_J/E_C > 2$. Additionally, the JJ should have a reasonably high apparent critical current to accommodate the quantized steps in the $I-V$ curve. The quantization at other frequencies is shown in Supplementary Figs. 1 and 2.

The quantized current plateaus are sensitive to the amplitude of the applied MW, $I_{ac}$. The intensity plot in Fig. 4 shows modulation of the $dV/dI$ at the position of the quantized current $I_{dc}=2efm$ with $I_{ac}$. This modulation is dual to the modulation of direct Shapiro steps with $V_{ac}$.

Such a behaviour is typical for the synchronization of the external radiation with the coherent tunnelling effects[14,21]. Modelling of $dV/dI$ for plateaus with different $m$ is shown in Supplementary Fig. 3.

## Current quantization
To understand the theory of the coherent phase slips in a small JJ, we consider a relatively simple limit of the junction with a large capacitance, including the stray capacitance $C_S$, and assume that $E_J \geq E_C$. It is well known that in this case, the phase slips lead to the energy band formation in the form $E(q) = -E_S \cos(\pi q/e)$, where $q$ is the electric charge flowing through the junction and the tunnelling energy, or the width of the bands, can be written as

$$E_S = \sqrt{\frac{8\eta}{\pi}} E_p e^{-\eta}, \qquad (2)$$

where $E_p = \sqrt{8E_J E_C}$ is the plasma energy of the JJ and $\eta = E_p/E_C = \sqrt{8E_J/E_C}$ (in our experiments $0.7 < E_J/E_C < 8.3$ and $2.4 < \eta < 8.2$)[1]. For the analysis, it is important to know that Eq. (2) works reasonably well even when $E_J > E_C$.

At non-zero bias current the dynamics of charge is affected by the impedance attached to the JJ, refer to Fig. 1b. In our circuit, the JJ is connected to an inductance $L$ and resistor $R$ so that $q(t)$ can be obtained from the Eq. (1):

$$L\ddot{q} + R\dot{q} + V_C \sin\frac{2\pi q}{2e} = V_{dc} + V_{ac}\cos(2\pi f t) + \xi(t). \qquad (3)$$

Here $V_{dc}$ is the $dc$ component of the applied voltage, $V_{ac}$ is the microwave signal, $V_C = \pi E_S/e$ is the critical voltage, and $\xi(t)$ is the noise. The normalised charge $2\pi q/2e$ is the dual analogue of the superconducting phase $\phi$ in the classical Josephson effect. At sufficiently strong noise $\xi(t)$, which translates to $E_S < k_B T$, the $I-V$ curve takes the form[3]:

$$V(I_{dc}, I_{ac}) = \sum_m J_m^2\left(\frac{I_{ac}}{2ef}\right) V_0(I_{dc} - 2efm) \qquad (4)$$

In the equation $J_m(x)$ are Bessel functions and $I_{dc}, I_{ac}$ are the $dc$ and $ac$ components of the current through the JJ, $I(t) = I_{dc} + I_{ac}\cos(2\pi f t)$. The equation describes the quantization of current with plateaus at $I_{dc} = 2efm$. The plateaus follow the square rather than the first order of the Bessel functions. It is a result of the operation in a regime of strong noise. One would have the first order of the Bessel function in the opposite case when $E_S > k_B T$. The $V_0(I_{dc})$ in the Eq.(4) is the $I-V$ curve measured without the microwave signal. It can be approximated with

$$V_0(I_{dc}) = \frac{V_C^2}{4R} \frac{I_{dc}}{I_{dc}^2 + \delta I_T^2}, \qquad (5)$$

for a thermal current noise $\delta I_T = \sqrt{k_B T/L}$[3]. We fit the quantized current steps in Fig. 3 using this equation with the thermal noise current $\delta I_T \approx 1$ nA. It should be noted that the noise is relatively large, $\delta I_T$ is comparable with $I_{ac}$.

## Analysis of the JJ parameters
To start the analysis we make a few comments. It is customary to characterize JJs by their critical current, $I_C = 2\pi E_J/\Phi_0$. However, in experiments, the apparent (measured) critical current, $I_C^*$, is usually smaller due to noise or other effects. We believe that in our sample $I_C^*$ is determined by the Landau–Zener tunnelling effect (Eq. (1)). For the critical voltage, $V_C = 2\pi E_S/2e$, the CQPS energy $E_S$ depends on both $E_J$ and $E_C$. The latter is calculated by taking the total capacitance, which includes $C_J$ together with the parasitic stray capacitance $C_S$. The

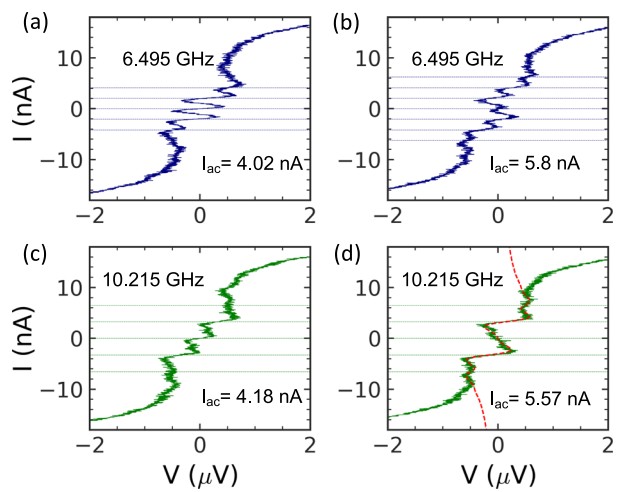

**Fig. 3 | The current quantization at the $I-V$ curves under MW radiation. (a, b)** $f = 6.495$ GHz, and **(c, d)** $f = 10.215$ GHz. The dashed lines correspond to the currents $I_{dc} = 2\,efm$, $m = 1,2,3,...$ The amplitude of MW is given as $I_{ac}$. The red dashed curve in **(d)** is the theoretical fit made with Eq. (4) (the thermal noise is taken as $\delta I_T = 1$ nA). The resistance at $m = 1$ plateau of this curve is 2.2 kΩ.

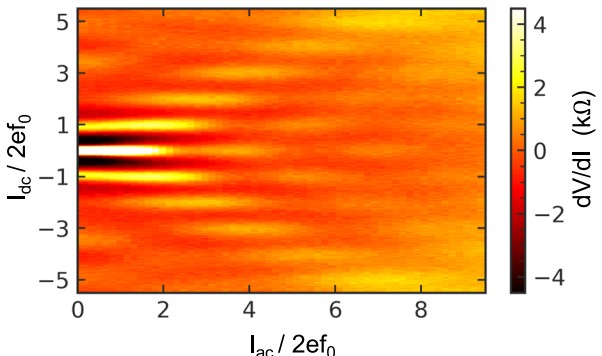

**Fig. 4 | An intensity plot of $dV/dI$ in coordinates of the normalised $dc$ and $ac$ current taken at 6.495 GHz.** The extremes of the d$V$/d$I$ correspond to the $I_{dc} = 2\,efm$ plateaus. They oscillate with MW amplitude $I_{ac}$.

apparent critical voltage $V_C^*$, observed in our experiments is, however, smaller than the calculated $V_C$. We attribute this to the effect of thermal noise.

To realise dual Shapiro steps in JJs one has to carefully tune the parameters of the JJs. We study several devices and find the parameter range, where the MW response and the current steps are present, see Fig. 5. The apparent critical voltage, $V_C^*$, is shown as a function of JJ area for different sets of $E_J$ and $E_C$. In all these measurements, the rest of the circuit is kept unchanged. We experimentally find that current quantization is seen in JJs with $E_J/E_C \geq 2$, $V_C^* \in [0.5\,\mu V, 5\,\mu V]$, and $I_C^*$ above 10 nA. By translating $E_C$, $E_J$, $V_C^*$, and $I_C^*$ to the parameters of the JJs one can find that the junction size should be of the order of $10^4$ nm$^2$ (e.g. $100 \times 100$ nm$^2$). We measure three sets of JJs with different pressures of oxygen during the growth of the insulating layer on Al (oxidation time is fixed to 10 min) - magenta, green, and brown data correspond to 1 μbar, 40 μbar, and 1 mbar oxidation pressures. They have low, medium and high $r$, correspondingly. Each set has its specific normal resistance $r = R_J A_J$, where $A_J$ is the junction area in units of nm$^2$.

The data are plotted with solid circles to indicate the measurements, in which the MW response is observed (either through direct step observation or in differential resistance). Crosses show samples where the MW response was not observed. The light red and blue areas

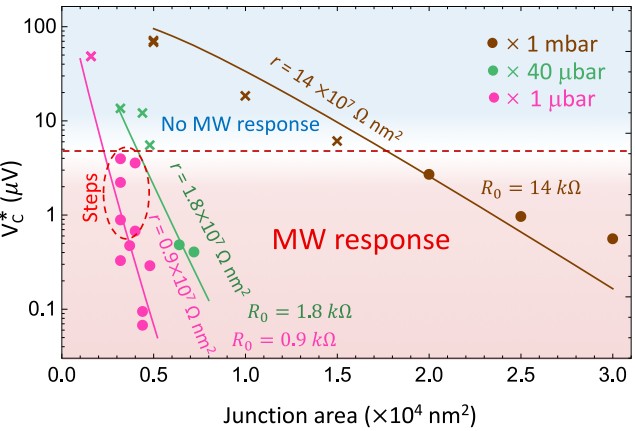

**Fig. 5 | Summary of measured devices.** Experimental $V_C^*$ versus the JJ area. Three groups of JJs (magenta, green, and brown circles) have different insulating layers with different specific resistances. The solid lines are fitted with Eq. (6). The dashed red line separates devices with and without explicit MW response. The devices demonstrating current quantization are inside the red dashed oval indicated as "Steps". The characteristic resistance $r$ expressed in units $10^7$ Ω × nm$^2$. The resistances for the reference area of $100 \times 100$ nm$^2$ are $R_0 = 0.9$ kΩ (magenta), 1.8 kΩ (green) and 14 kΩ (brown).

separate the data with visible responses to the MW radiation from those without responses. The separation is shown schematically by a red dashed line with $V_C^* \sim 5$ μV. We also show a narrow area by a dashed oval where the steps are observed directly (specified by "Steps" in the figure). All the steps are observed on the set of samples with the lowest specific resistance $r$, oxidized at 1 μBar. Also, note that the response (with direct steps or through $dV/dI$ measurements) is observed only for devices with apparent critical current $I_C^*$ above 10 nA. Examples of $I-V$ curves of samples with different oxidation parameters are shown in Supplementary Fig. 4.

To explain the data we derive $V_C^*$ in the limit of strong noise, $k_B T \geq eV_C$, where $T$ is the effective temperature of the system (e.g. resistors or equivalent noise coming from the outside circuit). If we assume that the $I-V$ curve is described by Eq. (5), then

$$V_C^* = \frac{V_C^2}{8R\delta I_T}. \tag{6}$$

We fit each set of our data in log-scale by Eq. (6), using the specific resistance $r$ as the fitting parameter. The results of fitting are shown as lines. The specific resistances are 0.9, 1.8, and $14 \times 10^7$ Ωnm$^2$, which correspond to 0.9, 1.8, 14 kΩ for the reference junctions of $100 \times 100$ nm$^2$ area. These values are close to the experimental values of 0.6, 2, and 15 kΩ taken with the witness JJs.

**Perspective**

The current quantization plateaus have slopes of $dV/dI$ below ~3 kΩ. The JohnsonNyquist noise of 6.3 kΩ Pd resistors can be one reason for this. Another possible reason is the heating of the chip with MWs. In the current design, the MW signal effectively resonates and heats the whole area around the JJ, with only a small portion of the MWs being coupled to the JJ. Erdmanis and Nazarov suggest bringing the MW through the gate electrode to a small island of the split JJ[17]. Such coupling can be highly efficient, allowing the use of a much weaker MW drive. In such a design, one can also modulate $E_S$ with the gate voltage $E_S = E_{S0}|\cos(\pi C_G V_G/e)|$. However, the effect of fluctuating charge around the split JJ may pose an issue.

Heating also unavoidably occurs in Pd resistors due to the $dc$ and $ac$ currents. Heating from $ac$ current can be minimized, but $dc$ current

cannot be avoided. For example, a *dc* current of 10 nA produces a load of about 10 pW. This results in the resistor heating to $T \sim 0.2$ K, or $k_B T/e \sim 10$ μV[22]. Simple solutions such as moving resistors to a separate chip with better heat dissipation[9] may not help, because, this increases the stray capacitance. Immersion cooling in the liquid He[3] is another approach to reducing noise[23], but the effect may be limited because the temperature in a system with phonon cooling weakly depends on the dissipated power

$$W : T \sim W^{1/5}.$$

Another way to improve quantization lies in increasing the apparent critical voltage. Partially this can be done by reducing the stray capacitance, and, simultaneously increasing the Josephson energy by reducing the $R_N$. In the regime of a larger blockade ($k_B T \ll eV_C^*$), the noise contribution will be exponentially smaller. With this approach, one would also have a larger $I_C^*$ needed for plateau observation at a higher current.

In summary we demonstrate current quantization (dual Shapiro steps) in small JJs and find the conditions for which the steps are observed. The maximum quantized current observed in our experiment reaches 7.7 nA, when MWs of 23.895 GHz are applied. The quantization is a result of the synchronization of the Bloch oscillations with the MW radiation. The coherent nature of the effect is confirmed by the observation of the modulation of the current plateau width with the MW amplitude. We show that the steps are observed in a limited range of the JJ parameters: junction area of about $0.3 \times 10^4$ nm², $R_0 \sim 0.6$ kΩ per $100 \times 100$ nm², $eV_C^* = 0.5$–5 μV. The current plateaus, however, are not flat, having slopes of $\sim 3.4$ kΩ at best. An advantage of the JJ devices, compared to those based on the nanowires, is their high yield. We suggest a few ways to optimize the devices to improve the accuracy and the amplitude of quantized currents. One approach is to reduce the stray capacitance in the circuit.

## Methods

The experimental samples are fabricated in four steps. We start with 5 nm thick TiN film on a Si wafer. The Ti/Au (10 nm/80 nm) macroscopic contact pads are fabricated with standard UV lithography. Electron Beam Lithography (EBL) is used during the next three steps: fabrication of compact TiN inductors (100 nm wide wire meanders) with negative resist and reactive ion etching in CF₄ plasma; deposition of compact resistors (15 nm thick and 150 nm wide Pd wire meanders) with the lift-off resist mask; fabrication of Al JJs using the shadow evaporation technique. In-situ ion milling is used to ensure good galvanic contact between the layers of the circuit.

Low-temperature experiments are conducted in a dry dilution refrigerator with a base temperature of 15 mK. Samples were mounted on a PCB with the DC and coplanar waveguide lines. The PCB with a sample is enclosed in a copper box. To suppress high-frequency noise *dc* lines pass through copper powder filters at room temperature, the thermo-coaxial cables, and low pass filters at low temperatures. The microwave signal is attenuated at various temperature stages of the refrigerator. The $I - V$ and $dV/dI$ curves are taken using the differential amplifier kept at room temperature[24]. An electric circuit for the amplifier is shown in Supplementary Fig. 5. $dV/dI$ curves are measured with a standard lock-in technique.

The height of zero bias peaks of $dV/dI$ (blockade of the JJ current) depends on the MW current amplitude as a square of the Bessel function of zero order. A significant peak suppression under particular frequencies indicates good coupling of the MW to JJ, refer to Supplementary Fig. 6. We use frequencies with good coupling to demonstrate the current quantization, see examples in Fig. 3. Coupling to MW drastically reduces above 30 GHz due to the high-frequency cut-off of our transmission lines. We explore the modulation of the current

quantization by the microwave amplitudes to prove the coherent nature of the effect, see Fig. 4.

## Data availability

The data generated in this study have been deposited to the Open Science Framework repository. They can be obtained without any restriction at https://osf.io/g4x9p/. Additional information, experimental curves and schemes are also provided in the Supplementary Information.

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

## Acknowledgements
This work was supported by Engineering and Physical Sciences Research Council (EPSRC) Grant No. EP/T004088/1, European Union's Horizon 2020 Research and Innovation Programme under Grant Agreement No. 862660/Quantum E-Leaps and 20FUN07 SuperQuant.

## Author contributions

O.V.A., R.S.S., and V.N.A. conceived and supervised the experiments. R.S.S., K.H.K., J.D., I.A. contributed to device fabrication and characterization at low temperatures and microwave ranges. All authors contributed to the simulation and analysis of the data. D.G. and V.N.A. wrote the manuscript and all authors contributed to editing the manuscript.

## Competing interests
The authors declare no competing interests.
