## [Peer Review File · Nature Communications]

REVIEWER COMMENTS

Reviewer #1 (Remarks to the Author):

In this work, the authors studied the microwave responses of a Josephson junction imbedded in a high impedance environment, and observed the current steps arising from the synchronization of the Bloch oscillations. They also explored the parameter space for observing these steps. The improvement of the current work is the replacement of a superconducting nanowire [Nature 608, 45 (2022)] with a Josephson junction. I found that this is a nice work and contributes to the understanding of the dual Shapiro steps. The following issues should be addressed before the recommendation of publication.

1. Eq. (3) is mentioned several times in the Results section, but it actually appears in the Discussion part. The presentation should be improved.
2. The microwave is applied through C_k , but also through L_2 . The inductance here seems to block the microwave signal. Is this the case? What is the effect of L_2 on the microwave?
3. Two frequencies, 6.495 GHz and 10.215 GHz, are used. Why these two specific frequencies are chosen? How about using frequencies a little bit away from these values, say 6.49 or 10.21 GHz? Why not use, for example, 6.5, 6, 10, 10.2 GHz?
4. Fig. 1(a) needs to be explained explicitly.
5. The authors measured many devices, and summarized the results. I believe the data should be plotted and presented in the supplementary. This point is particularly important. I suggest the authors to show many (at least typical) plots for different devices in different regimes.
6. Please add the information of current-driven or voltage-driven measurement.

Reviewer #2 (Remarks to the Author):

In their paper, Shaikhaidarov et al. report the observation of quantized current steps originating from the synchronization of Bloch oscillations with a strong microwave drive in circuits consisting of a small Josephson junction embedded in a high-impedance environment. This effect is usually referred to as dual Shapiro steps. This is an extremely important observation since this effect was predicted almost 40 years ago by Averin, Zorin and Likharev and it is only recently that convincing observations of this effect have been reported.

To realize this tour de force, the authors embedded a small Josephson junction in a high impedance environment consisting of a combination of inductances made of TiN meanders and Pd resistors. These resistances and inductances are organized in a four-probe configuration, allowing the measurement of the DC properties of the junction. They can also irradiate the Josephson junction with microwaves using

a capacitively coupled transmission line. This experimental configuration is a direct follow-up of the work done by a similar team (Shaikhaidarov et al., Nature, 2022) in which the authors also reported the observation of dual Shapiro steps but using a superconducting nanowire instead of a Josephson junction.

The authors first report a very enlightening $I(V)$ curve where both a Coulomb blocked and a supercurrent can be observed at the same time, which is very interesting in itself as also pointed by the authors.

When the device is irradiated with microwaves, the $I(V)$ curve develops clear current plateaus which are quantified as $2e \times \text{frequency} \times \text{an integer}$. The width of these plateaus varies with the applied microwave power, as predicted by theory.

Shaikhaidarov et al. also report a very interesting 'phase diagram' which describes the behaviour (existence or non-existence of a MW response) of the twenty samples they measured, covering a wide range of parameters. This figure is very valuable as it will guide the community in the design of future experiments on a similar subject.

The authors also provide a comparison between their data and theory. Most of the theory was already developed in their nanowire paper of 2022 but they extend it to provide a simple expression of the effective critical voltage V_c^* that allows them to explain nicely their phase diagram.

Overall, I believe that the results are solid and represent an important advance in the field of quantum electrical metrology, since the observation of dual Shapiro steps has posed a challenge to generations of condensed matter physicists.

However, I cannot recommend publication of this paper in its current form for two reasons. Firstly, there are too many typos and inaccuracies, which do a disservice to this important work (I will detail this in the second part of my report). Secondly, the authors barely comment on the work of Crescini et al. (reference [7] of the current project) and make no mention of the work of Kaap et al. (<https://arxiv.org/abs/2401.06599v1>) which has been available on the arXiv for over two months now. In my opinion, the fact that two other independent groups have observed dual Shapiro steps in small Josephson junctions is actually a very strong point for the work of Shaikhaidarov et al. It would be useful to comment on this work and clearly explain the differences, so that the community is aware of other observations of these dual Shapiro steps that have been eluded for decades since the seminal work of Averin, Zorin and Likharev. For example, the authors refer to the work of Crescini et al. as an alternative system with "controllable classical charge transfer". I think this is an important statement and the authors should clarify what they mean by this so that the community can compare the two works and understand the differences.

Now I would like to come back to more detailed comments on the current draft:

- Overall, there are many typos and erroneous wording, a proper revision of the draft would be welcome.
- Abstract: "Experimental confirmation of this phenomenon was delayed for a long time until last year's breakthrough when the current quantisation was demonstrated in the superconducting nanowires." This nice work was published in 2022, hence two years ago.
- Page 2: "we protect the JJ from the environment by embedding it in the electric circuit with the

inductance, normal resistors, and quasiparticle traps". These traps seem to be a key ingredient but I couldn't find any explanation about them in the paper. How do they work? How are they built? What are the requirements? The authors should comment that.

- Page 2: "the normal resistance of the JJ in combination with the protective circuit would be higher than the resistance quantum". If I look at the red points of figure 5, there are junctions with an area of ~ 0.5 times the reference area, which means that these junctions exhibit a normal resistance of around 2 kOhms. How can such a low resistance (that is directly in parallel with the Josephson tunneling element) combine with any resistance network to give a resistance larger than 6.5 kOhms?

- Caption figure 1 (b) do the authors mean SEM (Scanning Electron Microscope) and not FIB (Focused Ion Beam)?

- Page 4: the authors write " A DC four-probe measurement scheme has current, I_+ and I_- , and voltage, V_+ and V_- , leads passed through the low-pass filters". What are these low-pass filters? Do they appear on figure 1 (b)?

- Page 4: "Here, we take E_{CJ} determined mainly by the junction capacitance C_J , because for the frequencies of $f \sim I_C/2e$, the inductance provides high impedance and effectively isolates the junction from the rest of the circuit." Could the authors explain better what they have in mind? They seem to suggest that the inductances L_1 and L_2 decouple the Josephson junction from the capacitance C_s at high frequencies. The problem with this reasoning is that this stray capacitance is most likely before the inductances and not after. For example, the SEM picture of the JJ (central picture, figure 1 (b)) shows two large pads directly across the junction. According to the scale bar they are $\sim 1 \times 1 \mu\text{m}^2$ and could easily give rise to a stray capacitance of the order of 1 fF. Then the inductances will not help decoupling C_s from the Josephson junction.

- Page 5: "The peaks corresponding to the plateaus follow the square of the Bessel function.". Could the authors provide a cut of figure 4 and maybe a fit to illustrate that the data follow a Bessel function? This point is very important because the authors go on to state in the conclusion that this is irrefutable proof of the "coherent nature of the effect".

- Figure 5: The color code is misleading: Blue and Red represent 1mbar and 1mubar respectively but they also stand for No MW response and MW response.

- Page 7: "The apparent critical voltage, V_c^* , is shown...." It is the first time that the authors use the term "critical voltage". Later they use "blockade voltage". Readability would be improved by avoiding too many different names for the same concept.

- Page 9: "Another way to improve quantization lies in increasing the blockade voltage". As far as I understand Figure 5 suggests that a larger blockade voltage prevents the observation of MW response. So could the authors explain more clearly what they have in mind?

- Supplement: formula S.6 should probably read $V_c^* = V_c^2 / (8R \Delta I)$ and not $V_c^* = V_c^2 / (8R_J \Delta I)$

Reviewer #3 (Remarks to the Author):

Abstract:

- 1) “Experimental confirmation of this phenomenon was delayed for a long time until last year's breakthrough when ~~the~~ current quantisation was demonstrated in ~~the~~ superconducting nanowires.”
- 2) “Direct observation of current quantization in Josephson junctions (JJs) would address the fundamental question of Bloch oscillations and pave the way for practical metrological applications.”

Q: It's not clear what the term "fundamental question of Bloch oscillations" refers to or entails.?

- 3) “The current quantization exists, however, only in a narrow range of JJ parameters, specifically the critical current and capacitance.”

Q: In this study, the authors investigate how Josephson Junction (JJ) parameters affect the visibility of dual Shapiro steps. In the abstract, the authors highlight the critical current and capacitance as key parameters. However, throughout the manuscript, they interchangeably refer to different parameters, including E_J and E_C . Furthermore, in Figure 5, additional parameters such as V_C^* (apparent voltage), junction area, and specific normal resistance r are introduced, contributing to a lack of uniformity and clarity for the reader. It's important for the authors to establish a clear link between these parameters throughout the text to avoid confusion and ensure consistency in their analysis.

- 4) “The findings are important for fundamental physics and their application to quantum metrology.”

Introduction:

- 5) “However, the demonstration of the true current quantisation lay decades ahead, when the dual Shapiro steps were seen in superconducting nanowires[3]”

Comment: It is appropriate to add the pioneering work of J. S. Lehtinen, Zakharov, and K. Yu. Arutyunov. Coulomb blockade and Bloch oscillations in superconducting Ti nanowires. Phys. Rev. Lett. 109, 187001 (2012)

- 6) “The accuracy of the quantized steps is below the current steps demonstrated in the alternative system with controllable classical charge transfer.”

Q: Could you please clarify what you mean by “accuracy”? Does this refer to the measurements in nanowires? If so, please adjust the sentence as follows: “The accuracy of the quantized steps in these superconducting nanowires is below..”

Q: : Could you please clarify what “the alternative system” refers to? Make it explicit to increase readability. E.g. In respect to Ref .7: “the alternative system” is “an ultrasmall Josephson junction in a high impedance array of larger junctions.”

- 7) “The current quantisation due to the synchronisation of microwaves with the Bloch oscillation in JJ remained unanswered.”

Q: This sentence is ambiguous. What remains unanswered? Current quantisation is certainly already demonstrated in Ref.2.

8) “Finally, the yield of devices was below 30%.”

Q: It is unclear this refers back to the nanowires as the previous sentence in the text is related to JJs.

Conclusion: The structure of the first paragraph of the introduction requires significant improvement. It fails to provide an accurate and clear overview of the current state of the art and the problem statement. At this juncture, the scientific question they aim to address in this work remains unclear.

9) “In this work, we focus on the study of the dual Shapiro steps in the JJ.”

Q: Be more specific and indicate what is the novelty in comparison to other works. Dual Shapiro steps have been studied in JJs and demonstrated. To list a few:

- The impact of parasitic Capacitance was studied in: Lisa Arndt, Ananda Roy, and Fabian Hassler Phys. Rev. B 98, 014525 (2018)
- The effect of super inductances on dual Shapiro steps has been investigated in: W. Guichard and F. W. J. Hekking, Phys. Rev. B 81, 064508 (2010) and A. Di Marco, F. W. J. Hekking, and G. Rastelli Phys. Rev. B 91, 184512 (2015)
- There is also a recent publication on arxiv in this regard: Demonstration of dual Shapiro steps in small Josephson junction (arxiv.org) that demonstrates pronounced quantized current steps in small Josephson junctions by synchronizing their Bloch oscillations, to external microwave signals between 1 and 6 GHz. The authors demonstrate pronounced dual Shapiro steps and quantized currents up to 2 nA. While in this work the authors demonstrate dual Shapiro steps up to higher frequencies and up to 7.7nA it is a work that deserves attention.

10) “It is important to find parameters of the JJs and the surrounding circuit, which would enable to preserve of the Bloch bands from external current noise and, at the same time, to couple the microwaves to the JJ, see Fig. 1(a). The strongest challenge is to tackle Landau-Zener tunneling between the lowest and the higher Bloch bands, which facilitates frequent switching between the branches of the hysteretic IV curve.”

Q: Is this really the main challenge? Both sentences give an unclear message. The Landau-Zener tunnelling will be determined for a circuit with $E_J \sim E_C \sim \hbar \omega_p$ while the $E_S = eV_c/\pi$, the phase slip energy corresponding to the bandwidth of the lowest ground state Bloch band,

will have an exponential dependence on $e^{-\sqrt{\frac{8E_J}{E_C}}}$. Since the charging energies of parallel capacitances add up inversely, a parasitic capacitance will result in a strong increase of the total charging energy. This will result in an exponentially small phase-slip energy which in turn causes an exponential suppression of the amplitude of the Bloch oscillation. As such, I believe both issues are important: E_J and E_C must be chosen in a way to minimize Landau-Zener and maximise E_S . Meaning there will be a trade-off if you look at their dependence on both energy scales. Meaning you want a situation where the first and second band are already well separated but the energy of the lowest band still depends strongly on Q.

This sentence is also contradicting the last sentence of the conclusion “The main measure is to reduce the stray capacitance in the circuit.” Which would indicate that increasing the phase slip energy is the main issue to observe dual Shapiro steps and not Landau-Zener tunneling.

Besides the above problems, thermal and quantum noise can impact the observation of dual Shapiro steps strongly. Also, here the relevant energy scales must be considered. I think the authors can describe the situation in more detail, more instructive and indicate more in detail

which approaches have been explored to overcome these complexities in order to provide a clear and up-to-date state-of-the art for the reader.

Comment: In Figure 1 you can also indicate the bandgap for clarity and make it more pedagogical. Wouldn't it be more instructive to adjust the scale in units of E_C , indicate for which ratio E_J/E_C this figure is. Moreover, provide a clear and detailed description for the reader in the caption: E.g. "The band structure is $2e$ -periodic in Q ". "The phase-slip energy E_S corresponds to the bandwidth of the lowest energy band".etc. Likewise, the caption for Fig.1.b is incomplete and not well formulated: "Thin film TiN inductances L1 and L2 has the form of meanders."

- 11) "The work has two directions: we protect the JJ from the environment by embedding it in ~~the~~ an electric circuit comprising on chip inductors, resistors and quasiparticle traps, effectively creating a high impedance environment with the inductance, normal resistors, and quasiparticle traps; and optimise the parameters of the device so that the normal resistance of the JJ in combination with the protective circuit would be higher than the resistance quantum, R_Q , and the Josephson energy E_J is close to the charging energy E_C , i.e. for $E_J \sim E_C$ "

Comment: "directions" is bad terminology. The sentence must be split up in two readable shorter sentences. Moreover, here lies the novelty of the work. The protection from the environment by embedding it in a high impedance environment adding inductors, normal resistors and quasiparticle traps is nothing new and has been explored in other works. In the best of my knowledge, a detailed study on the JJs parameters is the novelty and this should be stressed. The structure must be improved.

- 12) "All these measures enable us to demonstrate ~~the~~ current quantisation in the JJs."

This sentence says nothing new at this point in the text. I suggest removing it completely.

- 13) "We observe ~~the~~ current steps up to $I_1 \sim 7.7 \text{ nA}$ ($= 2e \times 24 \text{ GHz}$). We prove also ~~the~~ coherent nature of the effect. ~~There is however a large field of research ahead: an increase of the quantised current above 10 nA, improvement of the quantisation accuracy, suppression of the noises (e.g. due to heating).~~ – this sentence is not good for the flow of reading. ~~Here~~ We present, in detail, data for one out of 20 devices demonstrating the current quantisation. All other devices fabricated with different parameters are analysed to determine the working regimes for the current quantisation."

Q: The authors aim to indicate what they will show in this manuscript. This includes: (i) the observation of current steps up to 24GHz (which the authors claim to be beyond state-of-the-art in JJ?) (ii) The coherent nature (How do you prove the coherent nature of the effect? Is the synchronisation between the Bloch oscillations and the external radiation not anyway a coherent effect?) (iii) that a similar analysis is performed on 20 devices with different parameters, specifically the junction size and square resistance of the junction. The regime to observe current quantisation with respect to the JJ parameters is investigated with respect to these two parameters.

It would be good for the manuscript to indicate already here the main conclusions of the work. The last point (iii) and the last sentence of the introduction is by no means a good way to end the introduction part and makes no bridge to the next part of the manuscript.

Results:

- 14) We also assume that there is a stray capacitance of $C_S \sim 1.2 \text{ fF}$ between the metallic parts of the circuit, which additionally shunts the junction at low frequencies.

Q: For an assumption this is a very precise value for the stray capacitance. Did you do any simulations? Why is this value selected?

- 15) The Pd/Al pads at both sides of LI serve as the quasiparticle traps.

Q: Can you provide any reference? How do they trap quasiparticles? Do you mean that it is an additional way to thermalize quasiparticles at base temperature? Why did you pick these values for the inductance and resistance? It should be clear that it is for creating a high impedance environment without too much local heating. Now the only info the authors indicate in this respect is the bandwidth of the circuit (which is probably of less importance)

- 16) The microwave (MW) signal is coupled to the JJ by the capacitor with $C_k = 0.1 \text{ fF}$.

Q: Why this value for capacitance? A reference to the methods section would be good here. How did you obtain a value for the amplitude of the rf current reaching the sample, I assume that the power reaching the sample depends in detail on the transfer function of the microwave line and the frequency?

Can you indicate in the caption of figure 2 that it is for JJ with no rf applied.

- 17) “From the experiment, we have a ratio..., which is close to that expected from the last equation when taking...”

Comment: To enhance readability, consider placing the equation for the Zener tunneling current on a separate line with an equation number for easy reference, rather than referring to it as "the last equation."

- 18) “The I-V curve drastically changes, when a microwave radiation (MW) is applied to the JJ: ~~the~~ current steps appear at...”

- 19) “At both frequencies the quantisation is limited by the amplitude of ~~the~~ I_C^* .”

- 20) Fig.3

Q: It appears that the observed steps do not precisely align with the expected current quantization steps. It would be good to mention the slope of the steps at this point in the text. Additionally, the steps should theoretically occur at integer values of $n \in \mathbb{Z}$ (I am curious if they only occur at integer values of $n \in \mathbb{Z}_0$). It would enhance clarity if the step numbers were indicated in the figure. Furthermore, where is the data displaying results up to 7.7nA? Given the significance emphasized by the authors in the abstract and introduction, it is crucial for them to explicitly present this result.

- 21) “We explore JJs with different E_J and E_C and find that the quantisation is present in a limited region of ratios $E_J/E_C > 2$. Also, the JJ should have a reasonably high apparent critical current to accommodate the quantised steps at the I-V curve.”

Q: At this point it is unclear what is meant with “should have a reasonably high apparent critical current”. As you indicate above the quantisation is limited by apparent critical current. This sentence is more related to figure 4 where you see that it is a more strict condition? As such, I do not think this sentence is in a good place in the manuscript as the results for different JJs are shown later in the text.

Comment:

The authors frequently reference equations such as Eq.3 before they are introduced in the text, which negatively impacts readability.

- 22) ~~“Such a behaviour this Bessel type dependence~~ is typical for the synchronisation of the external radiation with the coherent tunneling effects”

Reference: R. Pangotra, B. Raes, Clécio C. de Souza Silva, I. Cools, W. Keijers, J. E. Scheerder, V. V. Moshchalkov & J. Van de Vondel Communications Physics volume 3, Article number: 53 (2020) shows this very nicely for the classical Josephson effect. Maybe the authors can add this reference here.

Q: I am curious if the square of the Bessel function is coincidental. While it arises from the mathematical derivation, it represents the squared dependence observed in normal Shapiro steps (where the step widths are proportional to the Bessel functions). It seems that the power of the parameter governs this dependence. Could the authors provide insight on this matter?

Discussion:

- 23) Eq.3 uses subscript m while in the text n is used for the n th dual Shapiro step. I assume the index should run over all $n \in \mathbb{Z}$ not indicated in the text and equation. Moreover I think in Eq.3 $V(I_{dc})$ should be $V(I)$
- 24) In figure 5 it would be nice to also add an axes with E_J/E_C ratio or E_J and E_C separately
- 25) We also show a narrow area by a dashed oval where the steps are observed directly
Q: I'm uncertain about the meaning of the preceding sentence. How do you specifically define "directly"?
- 26) Q: The authors provide a clear explanation for the dependence of the data in Figure 5 but not a clear explanation why no MW response is observed above $V_C^* \sim 5\mu V$.

Perspective:

No comments

Conclusions:

- 27) “The coherent nature of the effect is confirmed by the observation of modulation of the current plateau width with the MW amplitude.”
Q: see point 13

28) Can you indicate why the yield in nanowire systems is so low?

Reviewer #1 (Remarks to the Author):

In this work, the authors studied the microwave responses of a Josephson junction imbedded in a high impedance environment and observed the current steps arising from the synchronization of the Bloch oscillations. They also explored the parameter space for observing these steps. The improvement of the current work is the replacement of a superconducting nanowire [Nature 608, 45 (2022)] with a Josephson junction. I found that this is a nice work and contributes to the understanding of the dual Shapiro steps. The following issues should be addressed before the recommendation of publication.

1. Eq. (3) is mentioned several times in the Results section, but it actually appears in the Discussion part. The presentation should be improved.

We removed the sentence referring to Eq. (3) as it is repeated in the discussion section. We only left the reference to Eq. (3) in the caption of Fig. 3 of the Result section. We also modified the text accordingly:

“The quantised current plateaus are sensitive to the amplitude of the applied MW, I_{ac} . The intensity plot in Fig. 4 shows modulation of the dV/dI at the position of the quantised current $I_{dc}=2ef m$ with I_{ac} . This modulation is dual to that of the direct Shapiro steps with V_{ac} . Such a behaviour is typical for synchronization of the external radiation with the coherent tunnelling effects [15, 16].”

2. The microwave is applied through C_k , but also through L_2 . The inductance here seems to block the microwave signal. Is this the case? What is the effect of L_2 on the microwave?

Microwaves can be delivered via capacitances C_k and inductances L_2 . The largest impedance comes from the coupling capacitance $|Z_{ck}|=|1/i\omega C_k| \sim 30 \text{ k}\Omega$ at frequency of 5 GHz, while the inductive impedance $Z_L=|i\omega L_2| \sim 100 \Omega$. Therefore, the inductance is not the limitation for the current. We can apply up to several millivolts to the coupling capacitance, therefore, $I_{ac} = V_{ac}/Z_{ck} \leq 100 \text{ nA}$.

3. Two frequencies, 6.495 GHz and 10.215 GHz, are used. Why these two specific frequencies are chosen? How about using frequencies a little bit away from these values, say 6.49 or 10.21 GHz? Why not use, for example, 6.5, 6, 10, 10.2 GHz?

The overall transmission frequency characteristic of our circuit is not flat. The coupling strength at different frequencies is identified by the lifting of the current blockade at low bias under the MW. We added the figure of dV/dI vs MW to the Supplement, which demonstrates the coupling strength. The 6.495 GHz and 10.215 GHz have good coupling to the JJ. We have compiled curves at different frequencies of different devices into one graph in the Supplement D.

4. Fig. 1(a) needs to be explained explicitly.

We added the description:

The Bloch bands are formed by the modulation of the system energy with the induced charge $q/2e$ with $2e$ periodicity, see the central panel of Fig. 1(a). The amplitude of this modulation, shown on the right panel of the figure is the CQPS energy, E_s . Three lowest Bloch bands are shown by different colours in the left panel. The system energy oscillates with the superconducting flux, Φ/Φ_0 . The energies in the graph are normalised by the JJ energy $E_J \Delta R_Q/2 R_N$, where R_Q and R_N are the quantum resistance and normal resistance of the JJ, Δ is the superconducting gap. The quantization of current happens when the coherent tunnelling of the system with $E(q/2e)$ is

synchronized with the external MW.

5. The authors measured many devices and summarised the results. I believe the data should be plotted and presented in the supplementary. This point is particularly important. I suggest the authors to show many (at least typical) plots for different devices in different regimes.

We added plots corresponding to different oxidation conditions and junction areas: Fig. S2 and S3 to Supplement D.

6. Please add the information of current-driven or voltage-driven measurement.

For the dc measurement we use a symmetric bias scheme with three instrumental amplifiers. There are two bias resistors $R_b=100$ kOhm in each arm of the scheme. The scheme operates in the voltage bias regime when sample resistance is low, $R \gg R_b$, and in the current bias regime in the opposite case. At the current plateau, we have $R < R_b$. This implies that it is an intermediate regime. This regime allows us to observe the "Bloch nose" at the I-V curve. The corresponding section with the scheme is added to the Supplement.

Reviewer #2

In their paper, Shaikhaidarov et al. report the observation of quantized current steps originating from the synchronization of Bloch oscillations with a strong microwave drive in circuits consisting of a small Josephson junction embedded in a high-impedance environment. This effect is usually referred to as dual Shapiro steps. This is an extremely important observation since this effect was predicted almost 40 years ago by Averin, Zorin and Likharev and it is only recently that convincing observations of this effect have been reported.

To realize this tour de force, the authors embedded a small Josephson junction in a high impedance environment consisting of a combination of inductances made of TiN meanders and Pd resistors. These resistances and inductances are organized in a four-probe configuration, allowing the measurement of the DC properties of the junction. They can also irradiate the Josephson junction with microwaves using a capacitively coupled transmission line. This experimental configuration is a direct follow-up of the work done by a similar team (Shaikhaidarov et al., Nature, 2022) in which the authors also reported the observation of dual Shapiro steps but using a superconducting nanowire instead of a Josephson junction.

The authors first report a very enlightening $I(V)$ curve where both a Coulomb blocked and a supercurrent can be observed at the same time, which is very interesting in itself as also pointed by the authors.

When the device is irradiated with microwaves, the $I(V)$ curve develops clear current plateaus which are quantified as $2e \times \text{frequency} \times \text{an integer}$. The width of these plateaus varies with the applied microwave power, as predicted by theory.

Shaikhaidarov et al. also report a very interesting 'phase diagram' which describes the behaviour (existence or non-existence of a MW response) of the twenty samples they measured, covering a wide range of parameters. This figure is very valuable as it will guide the community in the design of future experiments on a similar subject.

The authors also provide a comparison between their data and theory. Most of the theory was already developed in their nanowire paper of 2022 but they extend it to provide a simple expression of the effective critical voltage V_c^* that allows them to explain nicely their phase diagram.

Overall, I believe that the results are solid and represent an important advance in the field of quantum electrical metrology, since the observation of dual Shapiro steps has posed a challenge to generations of condensed matter physicists.

However, I cannot recommend publication of this paper in its current form for two reasons. Firstly, there are too many typos and inaccuracies, which do a disservice to this important work (I will detail this in the second part of my report). Secondly, the authors barely comment on the work of Crescini et al. (reference [7] of the current project) and make no mention of the work of Kaap et al. (<https://arxiv.org/abs/2401.06599v1>) which has been available on the arXiv for over two months now. In my opinion, the fact that two other independent groups have observed dual Shapiro steps in small Josephson junctions is actually a very strong point for the work of Shaikhaidarov et al. It would be useful to comment on this work and clearly explain the differences, so that the community is aware of other observations of these dual Shapiro steps that have been eluded for decades since the seminal work of Averin, Zorin and Likharev. For example, the authors refer to the work of Crescini et al. as an alternative system with "controllable classical charge transfer". I think this is an important statement and the authors should clarify what they mean by this so that the community can compare the two works and understand the differences.

We have acknowledged the work of Crescini et al. in the manuscript. We refer to the work as "controllable classical charge transfer" because no modulation of the quantised plateaus with the

MW amplitude has been demonstrated. Moreover, the observation is consistent with the operation of the single electron pump, which cannot be ruled out. To be more accurate we avoid the claim of classical transport in the new version of our manuscript and pasted the following neutral statement: The steps were also observed in a system with ultra-small JJs in a high impedance environment of large JJs [8] however the synchronisation (in Bessel oscillations) must be proven.

We were not aware of Kaap et al. at the time of submission of the manuscript to the Nature Physics on January 10 2024. Also added a paper of the same authors published on Jan 12 2024 in PRL The corresponding text and references are added.

“A similar demonstration of the synchronisation of Bloch oscillations with the MW and the current quantisation in small Josephson Junctions was reported recently by the PTB group [12, 13].

Now I would like to come back to more detailed comments on the current draft:

- Overall, there are many typos and erroneous wording, a proper revision of the draft would be welcome.

We did a proofreading of the text to correct the typos

Abstract: “Experimental confirmation of this phenomenon was delayed for a long time until lastyear's breakthrough when the current quantisation was demonstrated in the superconducting nanowires.” This nice work was published in 2022, hence two years ago.

We have corrected this by typing “in 2022”

- Page 2: “we protect the JJ from the environment by embedding it in the electric circuit with the inductance, normal resistors, and quasiparticle traps”. These traps seem to be a key ingredientbut I couldn't find any explanation about them in the paper. How do they work? How are they built? What are the requirements? The authors should comment that.

An explanation of resistors, inductances and quasiparticle traps is added to Supplement C.

- Page 2: “the normal resistance of the JJ in combination with the protective circuit would be higher than the resistance quantum”. If I look at the red points of figure 5, there are junctions with an area of ~ 0.5 times the reference area, which means that these junctions exhibit a normal resistance of around 2 kOhms. How can such a low resistance (that is directly in parallel with the Josephson tunneling element) combine with any resistance network to give a resistance larger than 6.5 kOhms?

Indeed, the experimental results show that the normal junction resistance is not limited by the quantum resistance. However, we do not see any contradiction, as in the working regime we do not exceed critical current and the junction does not switch to the normal regime.

The important parameter in a system is the differential quasiparticle resistance of the junctions at low bias, wherethe current quantisation occurs. Since the junctions are in the superconducting state, their differential resistance below the voltage gap $2\Delta/e$ is high, comparable with the quantum resistance. Although it is rather low in the normal state and at high bias voltage. We have modified the text accordingly.

- Caption figure 1 (b) do the authors mean SEM (Scanning Electron Microscope) and not FIB (Focused Ion Beam)?

Pictures are taken with He FIB. We have added “He” to the text.

- Page 4: the authors write “ A DC four-probe measurement scheme has current, I_+ and I_- , and voltage, V_+ and V_- , leads passed through the low-pass filters”. What are these low-pass filters? Do they appear on figure 1 (b)?

The low pass filters are not included in the scheme of Fig. 1. They are housed at the filter box at 15 mK. Each line has cascaded LTCC low pass filters with a stop band from 80 MHz to 20 GHz. The corresponding text is added to the manuscript

- Page 4: “Here, we take E_{CJ} determined mainly by the junction capacitance C_J , because for the frequencies of $f \sim I_C/2e$, the inductance provides high impedance and effectively isolates the junction from the rest of the circuit.” Could the authors explain better what they have in mind? They seem to suggest that the inductances L_1 and L_2 decouple the Josephson junction from the capacitance C_s at high frequencies. The problem with this reasoning is that this stray capacitance is most likely before the inductances and not after. For example, the SEM picture of the JJ (central picture, figure 1 (b)) shows two large pads directly across the junction. According to the scale bar they are $\sim 1 \times 1 \mu\text{m}^2$ and could easily give rise to a stray capacitance of the order of 1 fF. Then the inductances will not help decoupling C_s from the Josephson junction.

It is an important point. Indeed, we have two Al squares next to the JJ. The mutual capacitance of the Al pads is not completely negligible; however, it is below 1 fF. We estimate it to be about 0.12 fF, using calculations with FEM software and it is much smaller than the stray capacitance C_S (it follows from the difference in sizes of the pads and inductances). However, it can be accounted for in the Landau-Zener current estimates. Importantly, the Landau-Zener current gives an upper limit and an order of magnitude of the apparent current. To be more accurate, we now accounted this capacitance in the new version of our manuscript and the obtained Landau-Zener current, which is now even larger.

- Page 5: “The peaks corresponding to the plateaus follow the square of the Bessel function.”. Could the authors provide a cut of figure 4 and maybe a fit to illustrate that the data follow a Bessel function? This point is very important because the authors go on to state in the conclusion that this is irrefutable proof of the “coherent nature of the effect”.

We show modelling of the dV/dI under the MW for a frequency of 6.496GHz in Supplement. The model and fit parameters used are briefly explained.

- Figure 5: The colour code is misleading: Blue and Red represent 1 mbar and 1 mubar respectively but they also stand for No MW response and MW response.

We have changed the colours of the plots to magenta, green and brown.

- Page 7: “The apparent critical voltage, V_c^* , is shown....” It is the first time that the authors use the term “critical voltage”. Later they use “blockade voltage”. Readability would be improved by avoiding too many different names for the same concept.

We agree with the comment. Apparent critical voltage is introduced on page 4 in the section starting with “ A stationary...” as an experimental parameter. However later, on page 7, we used the wrong wording to introduce V_c^* again. We corrected the sentence on page 7.

- Page 9: "Another way to improve quantization lies in increasing the blockade voltage". As far as I understand Figure 5 suggests that a larger blockade voltage prevents the observation of MW response. So could the authors explain more clearly what they have in mind?

The accuracy of the quantisation should be improved in devices with larger V_C^* . But the MW response is indeed suppressed because of small I_C^* in samples with large V_C . To keep I_C^* high one can reduce R_N by making thinner the insulation layer of the JJ. We amended the text accordingly.

- Supplement: formula S.6 should probably read $V_C^* = V_C^2 / (8R \Delta I)$ and not $V_C^* = V_C^2 / (8R_J \Delta I)$

Corrected

Reviewer #3

We thank the referee for reviewing our manuscript. Below are our response to the referee's questions and comments.

Abstract:

1) "Experimental confirmation of this phenomenon was delayed for a long time until last year's breakthrough when the current quantisation was demonstrated in the superconducting nanowires."

corrected

2) "Direct observation of current quantization in Josephson junctions (JJs) would address the fundamental question of Bloch oscillations and pave the way for practical metrological applications."

Q: It's not clear what the term "fundamental question of Bloch oscillations" refers to orientails.?

We have revised this sentence: Direct observation of the current quantisation in the JJs would confirm synchronization of the Bloch oscillations with the MW and help for the realisation of the metrological current standard.

3) "The current quantization exists, however, only in a narrow range of JJ parameters, specifically the critical current and capacitance."

Q: In this study, the authors investigate how Josephson Junction (JJ) parameters affect the visibility of dual Shapiro steps. In the abstract, the authors highlight the critical current and capacitance as key parameters. However, throughout the manuscript, they interchangeably refer to different parameters, including E_J and E_C . Furthermore, in Figure 5, additional parameters such as V_C^* (apparent voltage), junction area, and specific normal resistance r are introduced, contributing to a lack of uniformity and clarity for the reader. It's important for the authors to establish a clear link between these parameters throughout the text to avoid confusion and ensure consistency in their analysis.

We thank the referee for bringing attention to this point.

To avoid possible confusion, we removed the part of the sentence "...the critical current and capacitance" in the abstract.

Indeed, our device is characterized by several parameters. The connection between them is shown in the text. However, to make it clearer, we added a paragraph with a summary of all parameters and explaining the link among them to the Discussion section:

First, we briefly summarize parameters, characterizing our device. Current transport through Josephson junctions is characterized by critical current $I_C = 2 E_J / \Phi_0$. However, in experiments the apparent (measured) critical current I_C^* is usually smaller due to noises or other effects. In our case, we suppose that I_C^* is determined by the Landau-Zener tunnelling effect and depends on E_J and the charging energy E_C of the JJ E_{CJ} is the charging energy determined by the effective JJ capacitance C_J (1). Similar for the CQPS effect in JJs, the critical voltage $V_C = 2\pi E_S / 2e$, where the CQPS energy E_S in JJ depends on the Josephson energy of the JJ and the charging energy E_C of the total capacitance, consisting of the junction capacitance C_J together with the parasitic stray capacitance C_S (the corresponding charging energies E_C and E_{CJ}). The apparent critical voltage V_C^* , observed in our experiments is smaller, we suppose, due to the thermal noise.

4) "The findings are important for fundamental physics and their application to quantum metrology."

Corrected

5) “However, the demonstration of the true current quantisation lay decades ahead, when the dual Shapiro steps were seen in superconducting nanowires [3]”

Comment: It is appropriate to add the pioneering work of J. S. Lehtinen, Zakharov, and K. Yu. Arutyunov. Coulomb blockade and Bloch oscillations in superconducting Ti nanowires. Phys. Rev. Lett. 109, 187001 (2012)

We added the reference but together with the relevant Comment on that paper in PRL: Also, note that the steps discussed in nanowires in Ref. [9] cannot be likely explained by superconducting behaviour as it is pointed out in Ref. [10].

6) “The accuracy of the quantized steps is below the current steps demonstrated in the alternative system with controllable classical charge transfer.”

Q: Could you please clarify what you mean by “accuracy”? Does this refer to the measurements in nanowires? If so, please adjust the sentence as follows: “The accuracy of the quantized steps in these superconducting nanowires is below..”

Q: Could you please clarify what “the alternative system” refers to? Make it explicit to increase readability. E.g. In respect to Ref .7: “the alternative system” is “an ultrasmlal Josephson junction in a high impedance array of larger junctions.”

We have revised the text accordingly:

The accuracy of the quantized steps recently demonstrated in nanowires is, however, below that of the alternative system with a controllable classical charge transfer through the quantum dot [6] and turnstile pumps [7].

7) “The current quantisation due to the synchronisation of microwaves with the Bloch oscillation in JJ remained unanswered.”

Q: This sentence is ambiguous. What remains unanswered? Current quantisation is certainly already demonstrated in Ref.2.

The Bloch waves appear in the periodic potential of a Josephson junction with a finite charging energy. This is essentially different from the nanowire, where the Bloch waves and geometrical charging energy are irrelevant.

8) “Finally, the yield of devices was below 30%.”

Q: It is unclear this refers back to the nanowires as the previous sentence in the text is related to JJs.

Conclusion: The structure of the first paragraph of the introduction requires significant improvement. It fails to provide an accurate and clear overview of the current state of the art and the problem statement. At this juncture, the scientific question they aim to address in this work remains unclear.

We have corrected: “A serious disadvantage for the nanowire devices is the fabrication yield, which is below 30%.”

9) “In this work, we focus on the study of the dual Shapiro steps in the JJ.”

Q: Be more specific and indicate what is the novelty in comparison to other works. Dual Shapiro steps have been studied in JJs and demonstrated. To list a few:

- The impact of parasitic Capacitance was studied in: Lisa Arndt, Ananda Roy, and Fabian Hassler Phys. Rev. B 98, 014525 (2018)
- The effect of super inductances on dual Shapiro steps has been investigated in

W. Guichard and F. W. J. Hekking, Phys. Rev. B 81, 064508 (2010) and A. Di Marco, F. W. J. Hekking, and G. Rastelli Phys. Rev. B 91, 184512 (2015)

- There is also a recent publication on arxiv in this regard: Demonstration of dual Shapiro steps in small Josephson junction (arxiv.org) that demonstrates pronounced quantized current steps in small Josephson junctions by synchronizing their Bloch oscillations, to external microwave signals between 1 and 6 GHz. The authors demonstrate pronounced dual Shapiro steps and quantized currents up to 2 nA. While in this work the authors demonstrate dual Shapiro steps up to higher frequencies and up to 7.7nA it is a work that deserves attention.

This is a very important point. None of these published papers, mentioned by the referee, studied the dual Shapiro steps experimentally because the effect has been observed only recently.

True demonstration of dual Shapiro steps in JJs is done by <https://arxiv.org/html/2401.06599v2>. By the time of submitting our manuscript to Nature Physics on January 10 2024 (which has been transferred to Nature Communications on Jan 30 2024) we were not aware of this work. Also, the work exists only as a preprint and has not passed the review process yet.

We added all the references, including the reference to the arxiv preprint as follows:
Different aspects of such a system have been previously studied theoretically [12–14].
and

A similar demonstration of the synchronisation of Bloch oscillations with the MW and the current quantisation in small Josephson Junctions was reported recently by the PTB group [18, 19].

10) “It is important to find parameters of the JJs and the surrounding circuit, which would enable to preserve of the Bloch bands from external current noise and, at the same time, to couple the microwaves to the JJ, see Fig.1(a). The strongest challenge is to tackle Landau-Zener tunneling between the lowest and the higher Bloch bands, which facilitates frequent switching between the branches of the hysteretic IV curve.”

Q: Is this really the main challenge? Both sentences give an unclear message. The Landau-Zener tunnelling will be determined for a circuit with $E_J \sim E_C \sim \hbar \omega_p$ while the $E_S = eV_C / \pi$ the phase slip energy corresponding to the bandwidth of the lowest ground state Bloch band, will have an exponential dependence on $\exp(-\sqrt{-8E_J/E_C})$
Since the charging energies of parallel capacitances add up inversely, a parasitic capacitance will result in a strong increase of the total charging energy. This will result in an exponentially small phase-slip energy which in turn causes an exponential suppression of the amplitude of the Bloch oscillation. As such, I believe both issues are important: E_J and E_C must be chosen in a way to minimize Landau-Zener and maximise E_S . Meaning there will be a trade-off if you look at their dependence on both energy scales. Meaning you want a situation where the first and second band are already well separated but the energy of the lowest band still depends strongly on Q.

We agree that there is a trade-off. However, achieving it in experiments may not be straightforward due to physical and technological limitations (E_J and E_C cannot be arbitrarily chosen) and possibly additional factors, which may not be accounted for in theory. We believe that finding the optimal parameters is indeed a big challenge otherwise the inverse Shapiro steps in nanowires (or elsewhere) would be demonstrated long time ago (see Q9). Our group spent many years and measured quite a few samples to find the steps, even though theoretically it was feasible, and we understood how to do it in principle.

We rephrased the statements.

This sentence is also contradicting the last sentence of the conclusion “The main measure isto reduce the stray capacitance in the circuit.”

Stray capacitance is parallel to C_J , which results in a decrease of E_C , and consequently, decrease of

E_S because of the exponential factor in Eqn. (1).

Which would indicate that increasing the phase slip energy is the main issue to observe dual Shapiro steps and not Landau-Zener tunneling. Besides the above problems, thermal and quantum noise can impact the observation of dual Shapiro steps strongly. Also, here the relevant energy scales must be considered. I think the authors can describe the situation in more detail, more instructive and indicate more in detail confidential which approaches have been explored to overcome these complexities in order to provide a clear and up-to-date state-of-the art for the reader.

Section Perspective answers the question. There is a limited range of JJ parameters to maximise the accuracy. We explored this range and found the optimal parameters of the JJ. To further improve the accuracy, one must tackle the noise. We suggest the directions to do this.

Comment: In Figure 1 you can also indicate the bandgap for clarity and make it more pedagogical. Wouldn't it be more instructive to adjust the scale in units of E_C, indicate for which ratio E_J/E_C this figure is.

The band diagram is calculated for the parameters of our model sample. The energy is normalised to E_J because E_J defines the potential and E_C has a meaning of the mass of the particle. We have added to the figure caption: "E_J/E_C = 5.5 of the model sample", and to the main text referring to the Fig. 1(f):

Moreover, provide a clear and detailed description for the reader in the caption: E.g. "The band structure is 2e-periodic in Q". "The phase-slip energy E_S corresponds to the bandwidth of the lowest energy band".etc.

We added the explanations to the main text.

Likewise, the caption for Fig1.b is incomplete and not well formulated: "Thin film TiN inductances L1 and L2 has the form of meanders."

We have corrected the caption: "... formed by TiN inductances L₁=1.15 μH and L₂=0.34 μH and normal Pd resistors R=6.3 kΩ"

11) "The work has two directions: we protect the JJ from the environment by embedding it in the an electric circuit comprising on chip inductors, resistors and quasiparticle traps, effectively creating a high impedance environment with the inductance, normal resistors, and quasiparticle traps; and optimise the parameters of the device so that the normal resistanceof the JJ in combination with the protective circuit would be higher than the resistance quantum, R_Q, and the Josephson energy E_J is close to the charging energy E_C, i.e. for E_J~ E_C"

Comment: "directions" is bad terminology. The sentence must be split up in two readable shorter sentences. Moreover, here lies the novelty of the work. The protection from the environment by embedding it in a high impedance environment adding inductors, normal resistors and quasiparticle traps is nothing new and has been explored in other works. In the best of my knowledge, a detailed study on the JJs parameters is the novelty and this should bestressed. The structure must be improved.

We have revised the structure of this section:

In response to the challenge, we explore the parameters of the JJ and environmental circuit to find a balance between maximising E_s and achieving adequate separation of the Bloch bands. At the same time, the JJ must have a differential quasiparticle resistance at low bias (below the

superconducting gap $2\Delta/e$, comparable to the resistance quantum, $R_Q = h/4e^2 \sim 6.5 \text{ k}\Omega$, and its Josephson energy close to the charging energy $E_J \sim E_C$ \cite{Averin1985, Erdmans2022}. Additionally, we protect the JJ from the EM noise of the environment by embedding it in an electric circuit with inductances, normal resistors, and quasiparticle traps (discussed in Supplement C).

12) "All these measures enable us to demonstrate the current quantisation in the JJs." This sentence says nothing new at this point in the text. I suggest removing it completely.

Removed

13) "We observe the current steps up to $I_1 \sim 7.7 \text{ nA}$ ($= 2e \times 24 \text{ GHz}$). We prove also the coherent nature of the effect. There is however a large field of research ahead: an increase of the quantised current above 10 nA, improvement of the quantisation accuracy, suppression of the noises (e.g. due to heating). – this sentence is not good for the flow of reading. Here We present, in detail, data for one out of 20 devices demonstrating the current quantisation. All other devices fabricated with different parameters are analysed to determine the working regimes for the current quantisation."

Q: The authors aim to indicate what they will show in this manuscript. This includes: (i) the observation of current steps up to 24GHz (which the authors claim to be beyond state-of-the-art in JJ?) (ii) The coherent nature (How do you prove the coherent nature of the effect?

Is the synchronisation between the Bloch oscillations and the external radiation not anyway a coherent effect?) (iii) that a similar analysis is performed on 20 devices with different parameters, specifically the junction size and square resistance of the junction. The regime to observe current quantisation with respect to the JJ parameters is investigated with respect to these two parameters.

It would be good for the manuscript to indicate already here the main conclusions of the work. The last point (iii) and the last sentence of the introduction is by no means a good way to end the introduction part and makes no bridge to the next part of the manuscript.

The coherent nature of our effect is proven by the Bessel-like oscillations. The demonstration of such oscillations is important to prove of coherent nature of the effect. Yes, the oscillations are observed in all our devices. Charge pumps represent the example of non-coherent devices. We have revised two sections to respond to the suggestions.

14) We also assume that there is a stray capacitance of $C_S \sim 1.2 \text{ fF}$ between the metallic parts of the circuit, which additionally shunts the junction at low frequencies.

Q: For an assumption this is a very precise value for the stray capacitance. Did you do any simulations? Why is this value selected?

The stray capacitance is found from the simulation of data in Fig.5 and confirmed by the numerical calculations.

We add more details:

We found from the simulation of the experimental points in Fig. 5 that there is an additional stray capacitance $C_S \approx 1.2 \text{ fF}$ between the junction and metallic parts of the circuit.

15) The Pd/Al pads at both sides of L1 serve as the quasiparticle traps.

Q: Can you provide any reference? How do they trap quasiparticles? Do you mean that it is an additional way to thermalize quasiparticles at base temperature? Why did you pick these values for the inductance and resistance? It should be clear that it is for creating a high impedance environment without too much local heating. Now the only info the authors indicate in this respect is the bandwidth of the circuit (which is probably of less importance)

The quasiparticle traps have been routinely used in superconducting nanostructures for a long time and are well-known by experimentalists (see for example APL 76.19 (2000): 2782-2784). We have added the reference to this work in the Supplement.

The quasiparticle trap is a galvanic contact with the material having a smaller superconducting gap. In our case, it is layered Al 70nm /Pd 15nm. Al has Δ smaller than TiN. Also, the Δ of Al is suppressed due to the proximity effect of the normal Pd film.

Values for inductance and resistance were found experimentally using available materials (TiN in our case) and we checked them in simulations using RSCJ-model. However, these values still have to be optimised. The bandwidth of the circuit is 0.6 GHz. It is given on page 4:

The screening circuit frequency band is $\Delta f_c = R/(2\pi L) = 0.6$ GHz.

16) The microwave (MW) signal is coupled to the JJ by the capacitor with $C_k = 0.1$ fF.

Q: Why this value for capacitance? A reference to the methods section would be good here.

How did you obtain a value for the amplitude of the rf current reaching the sample, I assume that the power reaching the sample depends in detail on the transfer function of the microwave line and the frequency?

We have designed $C_k = 0.1$ fF to be small to avoid additional shunting capacitance and to be able to deliver the required amplitudes of MW. Such coupling capacitance has been used and successfully worked in another experiment (e.g. Ref. [2]).

The value of I_{ac} is obtained from the simulation of the intensity graphs dV/dI vs I_{dc} and MW power by Eq. S13 of Supplement. An example of the simulations is in Fig. S3. We have added to the Supplement:

The simulation allows us to find the absolute value of I_{ac} which we use in Fig. 3 and 4.

Can you indicate in the caption of figure 2 that it is for JJ with no rf applied.

We have added to the figure caption: " without the MW ".

17) "From the experiment, we have a ratio..., which is close to that expected from the last equation when taking..."

Comment: To enhance readability, consider placing the equation for the Zener tunneling current on a separate line with an equation number for easy reference, rather than referring to it as "the last equation."

Implemented

18) "The I-V curve drastically changes, when a-microwave radiation (MW) is applied to the JJ: the current steps appear at..."

Corrected

19) "At both frequencies the quantisation is limited by the amplitude of the I_c^* ."

Corrected

20) Fig.3

Q: It appears that the observed steps do not precisely align with the expected current quantization steps. It would be good to mention the slope of the steps at this point in the text. Additionally, the steps should theoretically occur at integer values of $n \in \mathbb{Z}$ (I am curious if they only occur at integer values of $n \in \mathbb{Z}_0$). It would enhance clarity if the step numbers were indicated in the figure. Furthermore, where is the data displaying results up to 7.7 nA? Given the significance emphasized by the authors in the abstract and introduction, it is crucial for them to explicitly present this result.

A dashed line in Fig.3 indicates the quantised values of the current calculated for the corresponding frequencies. We add resistance at the $m=1$ plateau of 10 GHz curve.

We would not like to add numbers of current plateaus to the figure as it will make the figures overloaded.

The curve with a plateau at 7.7 nA ($f=23.895$ GHz) is shown in Fig. S1 of the Supplement. The effect at ~ 24 GHz was observed in two samples.

21) "We explore JJs with different E_J and E_C and find that the quantisation is present in a limited region of ratios $E_J/E_C > 2$. Also, the JJ should have a reasonably high apparent critical current to accommodate the quantised steps at the I-V curve."

Q: At this point it is unclear what is meant with "should have a reasonably high apparent critical current". As you indicate above the quantisation is limited by apparent critical current. This sentence is more related to figure 4 where you see that it is a more strict condition? As such, I do not think this sentence is in a good place in the manuscript as the results for different JJs are shown later in the text.

Comment:

The authors frequently reference equations such as Eq.3 before they are introduced in the text, which negatively impacts readability.

We have amended the text so that Eq. 3 (Eq. 4 in this version) is referenced only in the caption of Fig. 3 before it appears in the text.

22) "Such a behaviour this Bessel type dependence is typical for the synchronisation of the external radiation with the coherent tunneling effects"

Reference: R. Pangotra, B. Raes, Clécio C. de Souza Silva, I. Cools, W. Keijers, J. E. Scheerder, V. V. Moshchalkov & J. Van de Vondel Communications Physics volume 3, Article number: 53 (2020) shows this very nicely for the classical Josephson effect. Maybe the authors can add this reference here.

Added

Q: I am curious if the square of the Bessel function is coincidental. While it arises from the mathematical derivation, it represents the squared dependence observed in normal Shapiro steps (where the step widths are proportional to the Bessel functions). It seems that the power of the parameter governs this dependence. Could the authors provide insight on this matter?

The square of the Bessel function in the Eq. 3 (Eq. 4 in this version) is one fundamental difference between the direct and inverse Shapiro step. In the direct Shapiro steps, J^2 dependence indicates a loss of macroscopic quantum coherence, so-called the Daem steps. In the inverse Shapiro steps the J^2 is a consequence of the noise $\xi(t)$ in Eq.(2) (Eq. 3 in this version). The macroscopic coherence is still preserved. The Bessel function argument is still linear with $I_{ac}: I_{ac}/2ef$.

Discussion:

23) Eq.3 uses subscript m while in the text n is used for the nth dual Shapiro step. I assume the index should run over all $n \in \mathbb{Z}$ not indicated in the text and equation. Moreover I think in Eq.3 $V(I_{dc})$ should be $V(I)$

We corrected letters from n to m. $V(I_{dc})$ is corrected to $V(I_{dc}, I_{ac})$.

24) In figure 5 it would be nice to also add an axes with E_J/E_C ratio or E_J and E_C separately

We use V_C^* and JJ area in Fig. 5 as the data for samples with different parameters of JJ show systematic behaviour. Both parameters are the experimental ones. Since V_C^* is a non-linear function of both, E_C and E_J , it will be impossible to add an extra axis conceivably.

25) We also show a narrow area by a dashed oval where the steps are observed directly
Q: I'm uncertain about the meaning of the preceding sentence. How do you specifically define "directly"?

The effect of the MW can be seen in dV/dI using the lock-in technique. In the direct I-V curve, the steps may be hidden in the noise. Only samples encircled in an oval in Fig. 5 had clear quantisation steps at the I-V curve. All other samples had peaks of dV/dI at the position of quantised current $I_m=2efm$.

26) Q: The authors provide a clear explanation for the dependence of the data in Figure 5 but not a clear explanation why no MW response is observed above $V_C \sim 5\mu V$.

We do not have a clear explanation of this effect at the moment.

Perspective:
No comments

Conclusions:

27) "The coherent nature of the effect is confirmed by the observation of modulation of the current plateau width with the MW amplitude."

Q: see point 13

We believe that the quantisation of current at $I_m=2efm$ and modulation of the current plateaus with the MW power following the square Bessel function is the proof of the macroscopic coherent nature.

28) Can you indicate why the yield in nanowire systems is so low?

We give the answer in the Introduction section:

The fabrication of nanowires with a width of ~ 10 nm is a major challenge as it is at the limit of

modern nanotechnology. Even the nanowires of identical geometry have wide variations of the superconducting parameters, like critical current, and, consequently, the CQPS energy [8].

REVIEWER COMMENTS

Reviewer #1 (Remarks to the Author):

I would like to thank the authors for their efforts to address my concerns. Below are my responses. There are still some issues that should be addressed before the recommendation of publication.

1. OK.

2. The information of the impedance should be emphasized in the paper properly.

3-5. OK.

6. "The scheme operates in the voltage bias regime when sample resistance is low, $R \gg R_b$, and in the current bias regime in the opposite case." Is this correct? I also have concerns on the voltage or current bias regime. If one wants to measure Shapiro steps, voltage steps with a constant voltage over a range of current, current-bias mode is appropriate to probe the current range of the voltage plateau. In the opposite, if one wants to measure the dual Shapiro steps, current steps with a constant current over a range of voltage, voltage-bias mode is appropriate. "At the current plateau, we have $R < R_b$. This implies that it is an intermediate regime." Does the choice of the measurement method meet the scope? It seems to violate the main goal.

In addition, as pointed out by the other reviewers, there are many typos and inaccuracies, even in the rebuttal letter. What is ref. 2 in the supplement? The overall scientific rigor should be improved and guaranteed.

Reviewer #2 (Remarks to the Author):

The authors have taken all my comments and remarks into account. I am happy to recommend their work for publication.

Reviewer #3 (Remarks to the Author):

The authors have thoroughly addressed all my inquiries, resulting in substantial improvements in both the manuscript's clarity and its scientific merit. I want to congratulate them with this task.

The current version of the manuscript presents a clear study on dual Shapiro steps in a Josephson Junction (JJ), aligning with previous research by the same team, notably their 2022 publication on nanowires in Nature (referenced as Ref.3). The authors report the observation of dual Shapiro steps with currents up to 7.7nA and conduct a comprehensive analysis of the microwave (MW) response across various junction parameters. A critical finding is the identification of specific junction parameters under which current quantization is distinctly observed. The authors highlight the significant higher yield of observing these current steps in their devices, drawing a favourable comparison to the yield in nanowires.

I acknowledge the scientific accuracy and the intriguing nature of the findings and I therefore recommend the current form of this manuscript for publication in your journal.

Referee#1

2. The information of the impedance should be emphasized in the paper properly.

We have added to the main text on page 4 : “The capacitance makes the highest impedance for the MW at the operation frequency, $\sim 30 \text{ k}\Omega$ at 5 GHz, while the inductance L_2 between the C_k and the JJ has only $\sim 100 \text{ k}\Omega$ at this frequency. The role of L_2 is to screen the JJ from the noise of higher frequencies.”

3-5. OK.

6. “The scheme operates in the voltage bias regime when sample resistance is low, $R \gg R_b$, and in the current bias regime in the opposite case.” Is this correct? I also have concerns on the voltage or current bias regime. If one wants to measure Shapiro steps, voltage steps with a constant voltage over a range of current, current-bias mode is appropriate to probe the current range of the voltage plateau. In the opposite, if one wants to measure the dual Shapiro steps, current steps with a constant current over a range of voltage, voltage-bias mode is appropriate.

The question concerns the answers to Referee comments. Indeed there is wrong statement/typo in the response. We have amended the answer.

“For the dc measurement we use a symmetric bias scheme with three instrumental amplifiers. There are two bias resistors $R_b=100 \text{ k}\Omega$ in each arm of the scheme. The scheme operates in the voltage bias regime when sample resistance is *high*, $R \gg R_b$, and in the current bias regime in the opposite case.”

“At the current plateau, we have $R < R_b$. This implies that it is an intermediate regime.” Does the choice of the measurement method meet the scope? It seems to violate the main goal.

We have amended text to: “At the current plateau, we have $R \sim R_b$.”

The referee is correct that the ideal measurement scheme for the current plateau is the voltage bias with $R \gg R_b$. However we have gotten the best signal/noise ratio at $R \sim R_b$, the intermediate bias regime. It is related to the set of differential amplifiers used for the instrumental amplifier. The intermediate bias regime allows us also to observe the “Bloch nose” at the I-V curve, which is difficult to measure with the pure voltage bias. We are experimenting with the alternative pure voltage bias setups, which did not give any improvement so far.

In addition, as pointed out by the other reviewers, there are many typos and inaccuracies, even in the rebuttal letter. What is ref. 2 in the supplement? The overall scientific rigor should be improved and guaranteed.

We have added missing reference in the supplement and amended several typos in the text.

REVIEWERS' COMMENTS

Reviewer #1 (Remarks to the Author):

The corrections in the response letter should be reflected in the supplementary. There are still errors in the supplementary. After that, I recommend its publication.